

# Identification of primary genes in glomeruli compartment of immunoglobulin A nephropathy by bioinformatic analysis

Mohammed Khamis Miraji[*], Yichun Cheng[*], Shuwang Ge and Gang Xu

Department of Nephrology, Tongji Hospital affiliated to Tongji Medical College, Huazhong University of Science and Technology, Wuhan, Hubei, China

[*] These authors contributed equally to this work.

## ABSTRACT

The current study is aimed to explore the specific genes which are responsible for the manifestation of Immunoglobulin A nephropathy (IgAN). Gene expression profiles GSE37460, GSE93798 and GSE104948 were analyzed using biological informatics methods to identify differentially expressed genes (DEGs) in IgAN glomeruli samples which were then compared to normal control samples. Subsequently, the DEGs were overlapped to explore genes with significant expression in at least two profiles. Finally, the enrichment analysis was conducted and the protein-protein interaction (PPI) network was constructed for the overlapping DEGs. A total of 28 genes were up-regulated and 10 genes were down-regulated. The up-regulated genes including CD44 and FN1 were chiefly involved in extracellular matrix receptors interaction pathway. In addition, CX3CR1 and CCL4 were associated with chemokine signaling pathway. ITGB2, PTPRC, FN1, and FCER1G were hub genes with a high degree of interaction in the PPI network. Therefore, this study identified many significant genes associated with extracellular matrix expansion and inflammatory mechanism which may be the novel biomarker and target candidates in IgAN.

Corresponding authors
Shuwang Ge,
geshuwang@tjh.tjmu.edu.cn
Gang Xu, xugang@tjh.tjmu.edu.cn

## INTRODUCTION

Immunoglobulin A nephropathy (IgAN) is the most common type of biopsy-proven primary glomerulonephritis and highly prevails in Asia, especially in China where it accounts for 58.2% of the glomerulonephritis cases (*Donadio & Grande, 2002*). The outcome of IgAN is highly variable, 15∼40% of the patients with IgAN progressively develop to end-stage renal disease (ESRD) within 20 years after diagnosis (*Barbour et al., 2013*). Therefore, IgAN has been a health concern throughout the world.

Pathogenic steps of the IgAN include deposition of IgA immune complexes, mesangial cells proliferation, and over-production of extracellular matrix components and infiltration of the inflammatory cells in the kidney tissues. Although these pathogenic steps of IgAN have been studied for decades (*Wyatt & Julian, 2013*), their mechanism is still unclear.

**Table 1 Characteristic of datasets included in the analysis.**

| Series number | Tissue | Platform | IgA nephropathy | Health control |
|---|---|---|---|---|
| GSE37460 | glomeruli | GPL14663 | 9 | 17 |
| GSE93798 | glomeruli | GPL22945 | 20 | 22 |
| GSE104948 | glomeruli | GPL24120 | 27 | 3 |

Gene microarray analysis is a great technique to detect the expression of thousands of genes and become an important technology for identifying genes and biological pathways that associate with various diseases. This approach is useful for identifying potential diagnostic, prognostic and therapeutic biomarker and has been applied for gene expression profiling in human IgAN (*Cox et al., 2010*; *Liu et al., 2017*). In addition, the protein–protein interaction (PPI) network was used to explore the function of proteins and disclose the rules of cellular activities including growth, development, metabolism, differentiation, and apoptosis (*Szklarczyk et al., 2011*). Recognition of protein interaction in a genetic study is essential in understanding the cellular control mechanism of the proteins.

In order to reveal crucial candidate genes contributing to IgAN, we conducted a series of microarray analysis of three microarray datasets which were obtained from the Gene Expression Omnibus (GEO) database. The genes with different expression (DEG) between IgAN patients and normal subjects were identified, and the overlapping DEGs were selected to perform enrichment analysis and construct a protein–protein interaction (PPI) network.

## METHODS

### Data resources

We searched the GEO database (http://www.ncbi.nlm.nih.gov/geo/) with the keywords "IgA nephropathy" and "Gene expression profile". By January 24, 2019, a total of 22 datasets were considered. These microarray datasets were selected based on the criteria that the samples must be human glomeruli tissue and contain IgA nephropathy and healthy subjects. Finally, three datasets were retained for subsequent analysis (Table 1). The process of data collection and selection was provided in Fig. S1.

### Data processing and differential genes analysis

The raw data was collected in the format of a CEL file and an annotation file. The data was initially preprocessed for background correction and quantile normalization using the Robust Multi-array Average (RMA) algorithm in the Affy package (*Kauffmann, Gentleman & Huber, 2009*). Quality control was performed by using MetaQC package (*Wang et al., 2012*), which provides four quantitative quality control indexes, including internal quality control (IQC), external quality control (EQC), accuracy quality control (AQCg and AQCp) and consistent quality control (CQCg and CQCp). When multiple probes matched to the same gene, we adopted the "IQR" method to select a probe with the largest interquartile range of gene expression values among all matched probes to represent the gene.

### Identification of the overlap DEGs between three microarray datasets

The processed data was used to run the linear Model for Microarray (LIMMA) package in R (*Ritchie et al., 2015*). The model was applied to identify differentially expressed genes between IgAN patients and healthy subjects in each dataset. The multiple testing correction was carried out to control the false discovery rate (FDR) with the application of the Benjamini–Hochberg procedure (*Benjamini & Hochberg, 1995*). The genes with the corrected $p$ value <0.05 and $|\log_2 FC| > 1$ were considered as DEGs. Then we examined the overlaps of the top 100 ranked genes across the three series. Genes with significantly differential expression in at least two datasets were selected as common genes. In order to evaluate the reliability of the above DEGs, we also performed a meta-analysis by the Fisher method and the maximum $P$-value method in MetaDE package (*Wang et al., 2012*).

### Enrichment analysis of the overlapped DEGs

The function and pathway enrichment analysis of the overlapped differentially expressed genes were carried out using the online web resource Database for Annotation, Visualization and Integration Discovery (DAVID, http://david.abcc.ncifcrf.gov). DAVID was used to conduct Gene Ontology (GO) and Kyoto Encyclopedia of Genes and Genomes (KEGG) pathway analysis in up-regulated and down-regulated overlapping DEGs independently. The $p < 0.1$ and gene count (number of enriched genes in a specific function or a pathway) >2 were considered the significant threshold values for the GO terms and pathway terms enrichment in the DEGs.

### Construction of the PPI network

To understand the interactions of the overlapped DEGs at the molecular level, we constructed the PPI network using the Search Tool for the Retrieval of Interacting Genes database (STRING, http://string.embl.de/) (*Szklarczyk et al., 2011*). The protein pairs with a confidence score >0.6 were considered to be significant. PPI network was visualized using Cytoscape software (http://cytoscape.org/) (*Smoot et al., 2011*).

## RESULTS

### DEGs and overlap DEGs identified in three microarray datasets

After the raw data of the three datasets were normalized, quality control was further performed for the datasets (Table S1). All three datasets were included for the subsequent analysis. A total of 217, 5,399 and 564 genes from GSE37460, GSE93798 and GSE104948 dataset were differentially expressed between IgA nephropathy and health controls respectively. Twenty-eight up-regulated and 10 down-regulated DEGs were identified in at least two datasets based on top 100 DEGs in the overlapping analysis (Fig. 1). Comparatively, FN1 gene and ALB gene were commonly expressed in all three datasets for up-regulation and down-regulation profiles respectively (Table 2).

### Functional and pathway enrichment of overlapping DEGs

The DEGs were annotated in the GO database and assigned to three categories, including biological processes, molecular functions and cellular components (Fig. 2). Biological

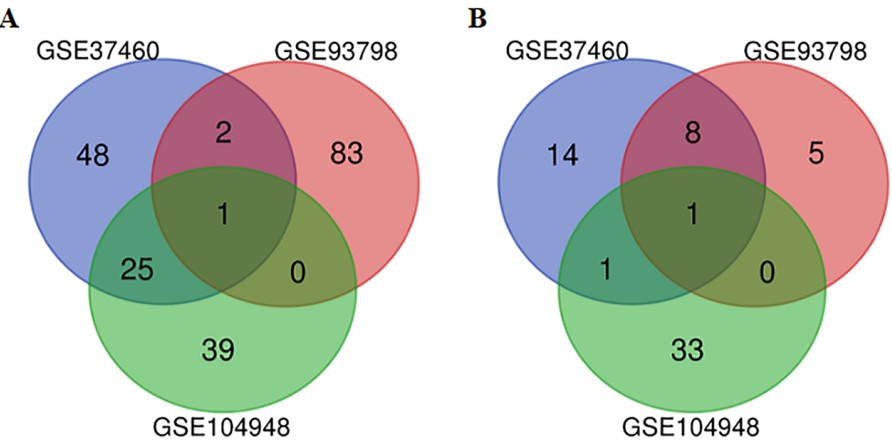

**Figure 1** A Venn diagram showing the differentially expressed genes in GSE37460, GSE93978, and GSE104948. (A) Up-regulated genes. (B) Down-regulated genes.

process enrichment analysis indicated that up-regulated DEGs were mainly involved in cell adhesion, leukocyte migration, and inflammatory response. Down-regulated DEGs were mainly involved in positive regulation of transcription from RNA polymerase II promoter, skeletal muscle cell differentiation and negative regulation of transcription from RNA polymerase II promoter. Cellular component analysis showed that up-regulated DEGs were mainly involved in the extracellular region, collagen trimer and extracellular space. Down-regulated DEGs were mainly involved in a nucleus. The molecular function analysis showed that up-regulated DEGs were mainly involved in protein binding receptor binding and collagen binding. Down-regulated DEGs were mainly involved in sequence-specific DNA binding, transcriptional activator activity, and DNA binding.

In addition, the up-regulated DEGs were significantly enriched in 12 pathways such as NF-kappa B signaling pathway including LYN, LY96, CCL4, CD14; ECM-receptor interaction pathway including CD44, COL6A3, COL1A2, FN1 and Amoebiasis pathway including COL1A2, ITGB2, CD14, FN1 (Table 3). No significant pathways were involved with down-regulated DEGs.

## PPI network of overlapping DEGs

The resultant network contained 29 nodes and 60 edges (Fig. 3). Furthermore, the overlapped DEGs such as ITGB2, PTPRC, FN1, and FCER1G in the PPI network were identified as hub genes.

## DISCUSSION

IgA nephropathy is the leading form of glomerulonephritis worldwide (*Wyatt & Julian, 2013*). Many studies have been conducted to explore the pathogenesis of IgAN (*Robert et al., 2015*; *Suzuki et al., 2011*), however, the mechanism underlying IgAN progression has not been fully elucidated. In this study, we identified candidate genes critical to IgAN by

**Table 2   Differentially Expressed Genes Identified in at least two datasets.**

| | Log$_2$FC | | | Meta-$P$-value | |
|---|---|---|---|---|---|
| | GSE37460 | GSE93798 | GSE104948 | Fisher | MaxP |
| FN1 | 2.32 | 3.58 | 2.23 | 1.0E-20 | 8.9E-06 |
| HBA1 | 3.50 | – | 3.53 | 1.0E-20 | 1.0E-20 |
| HBB | 3.32 | – | 3.34 | 1.0E-20 | 1.1E-04 |
| FCN1 | 1.99 | – | 2.85 | 1.0E-20 | 2.1E-01 |
| TYROBP | 1.81 | – | 2.95 | 1.0E-20 | 1.1E-01 |
| TIE1 | 1.71 | – | 1.92 | 1.0E-20 | 1.8E-01 |
| IGFBP5 | 1.71 | 3.92 | – | 5.9E-06 | 2.4E-03 |
| HCK | 1.69 | – | 2.42 | 1.0E-20 | 9.3E-01 |
| FCER1G | 1.68 | – | 2.87 | 1.0E-20 | 1.0E-20 |
| PTPRC | 1.64 | – | 3.14 | 1.0E-20 | 1.0E-20 |
| NETO2 | 1.61 | – | 2.08 | 1.0E-20 | 9.6E-04 |
| LY96 | 1.56 | – | 2.48 | 1.0E-20 | 7.6E-03 |
| CD14 | 1.55 | – | 1.90 | 1.0E-20 | 1.0E-20 |
| CD53 | 1.55 | – | 2.67 | 1.0E-20 | 1.7E-03 |
| CCL4 | 1.50 | – | 2.00 | 7.4E-07 | 3.1E-02 |
| ITGB2 | 1.50 | – | 2.59 | 1.0E-20 | 1.9E-01 |
| IFI30 | 1.49 | – | 1.74 | 1.0E-20 | 1.0E-20 |
| COL15A1 | 1.48 | – | 1.77 | 1.0E-20 | 1.0E-20 |
| COL1A2 | 1.44 | – | 2.77 | 2.2E-06 | 3.1E-01 |
| CX3CR1 | 1.42 | – | 3.45 | 1.0E-20 | 2.5E-01 |
| COL6A3 | 1.41 | – | 2.05 | 1.5E-06 | 1.0E-20 |
| CD44 | 1.40 | – | 3.12 | 1.0E-20 | 1.0E-20 |
| CTSS | 1.40 | – | 2.37 | 1.0E-20 | 3.1E-01 |
| LYN | 1.39 | – | 2.07 | 1.0E-20 | 3.0E-01 |
| TBX3 | 1.32 | – | 2.26 | 1.0E-20 | 3.3E-03 |
| POSTN | 1.32 | 3.47 | – | 7.4E-07 | 8.8E-05 |
| SAMSN1 | 1.31 | – | 1.75 | 7.4E07 | 1.0E-20 |
| CYBB | 1.30 | – | 2.28 | 1.0E-20 | 5.6E-02 |
| ALB | −1.67 | −3.42 | −2.39 | 4.4E-06 | 3.7E-05 |
| FOSB | −2.77 | −6.09 | – | 1.0E-20 | 3.2E-01 |
| NR4A3 | −2.10 | −3.76 | – | 1.0E-20 | 2.4E-02 |
| SIK1 | −2.07 | −3.42 | – | 1.0E-20 | 3.2E-02 |
| NR4A2 | −1.99 | −3.71 | – | 1.0E-20 | 1.3E-03 |
| ATF3 | −1.98 | −4.82 | – | 1.0E-20 | 7.5E-01 |
| MAFF | −1.71 | −3.62 | – | 6.0E-05 | 8.4E-02 |
| IGF1 | −1.60 | – | −1.82 | 1.0E-20 | 1.0E-20 |
| EGR1 | −1.44 | −4.19 | – | 5.6E-05 | 9.4E-01 |
| EGR3 | −1.29 | −3.32 | – | 5.0E-05 | 8.9E-06 |
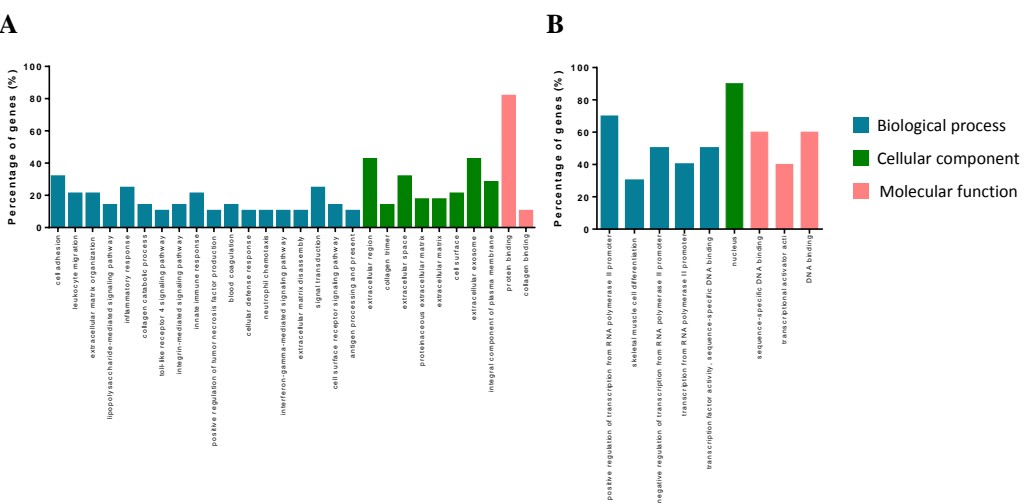

**Figure 2** **GO term of overlapped differentially expressed genes.** Blue bars, biological process; green bars, cellular component; red bars, molecular function. (A) Up-regulated genes. (B) Down-regulated genes.

**Table 3** **Pathways significantly enriched by up-regulated differentially expressed genes.**

| ID | Term | Count | *P* value | Genes |
|---|---|---|---|---|
| hsa04064 | NF-kappa B signaling pathway | 4 | 0.002 | LYN, LY96, CCL4, CD14 |
| hsa04512 | ECM-receptor interaction | 4 | 0.002 | CD44, COL6A3, COL1A2, FN1 |
| hsa05146 | Amoebiasis | 4 | 0.003 | COL1A2, ITGB2, CD14, FN1 |
| hsa05144 | Malaria | 3 | 0.009 | ITGB2, HBA1, HBB |
| hsa05152 | Tuberculosis | 4 | 0.014 | FCER1G, ITGB2, CTSS, CD14 |
| hsa04062 | Chemokine signaling pathway | 4 | 0.016 | LYN, HCK, CX3CR1, CCL4 |
| hsa05133 | Pertussis | 3 | 0.020 | LY96, ITGB2, CD14 |
| hsa04666 | Fc gamma R-mediated phagocytosis | 3 | 0.024 | PTPRC, LYN, HCK |
| hsa04974 | Protein digestion and absorption | 3 | 0.026 | COL6A3, COL1A2, COL15A1 |
| hsa04620 | Toll-like receptor signaling pathway | 3 | 0.037 | LY96, CCL4, CD14 |
| hsa04650 | Natural killer cell mediated cytotoxicity | 3 | 0.048 | FCER1G, ITGB2, TYROBP |
| hsa04611 | Platelet activation | 3 | 0.054 | LYN, COL1A2, FCER1G |
| hsa04145 | Phagosome | 3 | 0.069 | ITGB2, CTSS, CD14 |
| hsa05143 | African trypanosomiasis | 2 | 0.092 | HBA1, HBB |

combining three datasets from GEO database and presented the potential pathways that may play an important role in IgAN.

Our bioinformatic analysis demonstrated that fibronectin (FN1) is overexpressed in IgAN glomeruli. The fibronectin is an essential element of the extracellular matrix. In pathological conditions, fibronectin could act as a seed for the deposition of ECM proteins around somatic cells, leading to sclerosis or fibrosis of tissue (*Brotchie & Wakefield, 1990*). Besides, some animal experiments have shown that fibronectin is associated with the progression of kidney disease (*Kubosawa & Kondo, 1998*; *Shui et al., 2006*). In patients with glomerulonephritis, increased plasma and urinary fibronectin levels were observed

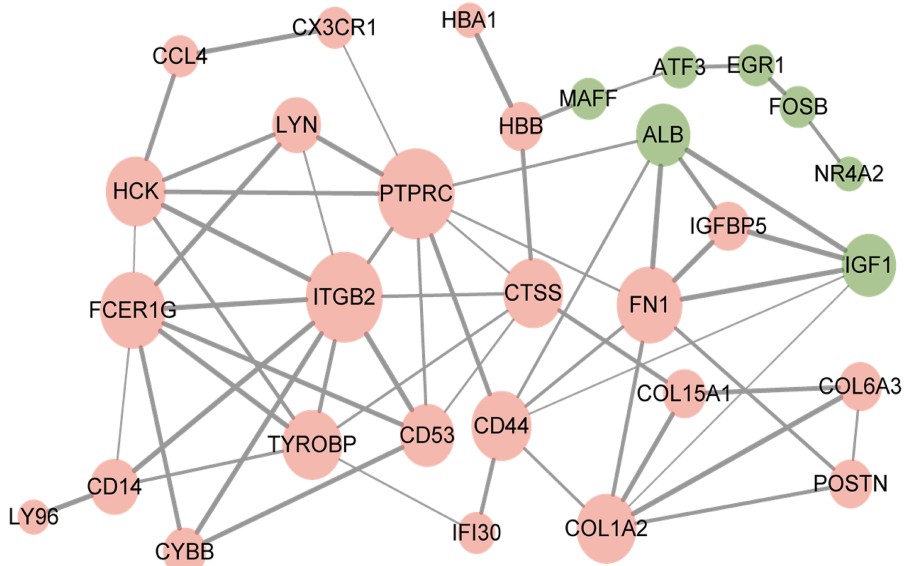

**Figure 3** **Protein–protein interaction network of overlapped differentially expressed genes.** Red circles, up-regulated genes; green circles, down-regulated genes.

(*Altunkova, Minkova & Belovezhdov, 1993*; *Idasiak-Piechocka et al., 2010*). The previous study also revealed that patients with IgAN have high circulating complexes containing Ag antibodies and Fibronectin (*Cederholm et al., 1988*), as it is in the preferred immune complex mechanism of primary IgA nephropathy. Therefore, the expression of FN1 gene may also influence the progression of IgAN, and thereby proper regulation of FN1 expression can effectively prevent IgAN.

Chemokines and chemokine receptors are potential in the homing and recruitment of specific immune cells. The existence of chemokine receptors expressing cells has been reported in patients with different types of nephropathies (*Zhuang, Cheng & Ming, 2017*). Our results demonstrate significant over-expression of chemokine (CX3-C motif) receptor 1 (CX3CR1) and chemokine (C-C motif) ligand 4 (CCL4) which also participate in the chemokine signal pathway. CX3CR1 is a transmembrane protein which is involved in the adhesion and migration of leukocytes and enhances the infiltration of cytotoxic lymphocytes (*Nishimura et al., 2002*), CX3CL1/CX3CR1 axis can initiate a cascade via several signaling pathways in the kidney, including ROS/MAPKS, Raf/MEK1/2-ERK1/2-Akt/PI3K and nuclear factor kappa (NF-kB) light polypeptide gene enhancer in B cells (*Zhuang, Cheng & Ming, 2017*). Also, it is reported that the CX3CL1/CX3CR1 axis can up-regulate mesangial cell expansion directly via Reactive Oxygen Species (ROS) and Mitogen-Activated Protein Kinase (MAPK) in diabetic nephropathy (*Park, Song & Ha, 2012*). The amount of glomerular and urinary fractalkine was higher in IgAN patients with recurrent episodes of gross hematuria compared with other patients with microscopic or no hematuria (*Cox et al., 2012*). Blocking the CX3CR1 with anti CX3CR1 antibody stops the migration of leukocytes into the glomeruli and prevents crescent formation and ameliorate

renal function (*Feng et al., 1999*), thus inhibiting deterioration of renal function to ESRD in IgAN. On the other hand, CCL4 is the chemokine with specificity for CCR5 receptors. The d32–CCR5 polymorphism played a significant role in the progression of primary IgAN, with the nl/nl genotype being an independent protective factor for late progression towards end-stage renal disease (*Berthoux et al., 2006*). Up-regulation of CCR5 is demonstrated in the kidney and its expression is related to the increase in proteinuria (*Navarro-Gonzalez et al., 2011*). Therefore, we believe that these chemokines and chemokine receptors may play an important role in the pathogenesis of IgAN and a large number of studies should be conducted to clear the mechanism in the future.

The PPI network shows that some hub genes have high interaction with other genes, comparatively. High interaction of these genes demonstrates their potential participation in the manifestation and progression of IgAN. ITGB2 gene is the most interactive hub gene in the network, and it is up-regulated, enriched in an inflammatory response and cell adhesion. ITGB2 gene is on chromosome 21 (21q22.3) and encodes integrin β2 protein (CD11b/CD18) (*Yassaee et al., 2016*). Upon inflammatory stimuli, CD11b/CD18 is rapidly activated via a conformational switch to mediate leukocyte migration from circulation to the inflamed tissue by binding to ICAM-1 (*Hynes, 2002*). Recent studies revealed that inhibition of the CD11b/CD18 could prevent the acute kidney injury and the progression of acute kidney injury to chronic kidney disease (*Dehnadi et al., 2017*; *Yago et al., 2015*). Of note, the infiltration of the inflammatory cell is an important characteristic in IgAN (*Wyatt & Julian, 2013*). Therefore, inflammatory response and cell adhesion via CD11b/CD18 may have a significant effect in IgAN pathogenesis initiation.

The observation of the ALB gene in the PPI network is peculiar. The ALB is encoding albumin which is chiefly found in urinary protein. The ALB possesses high connectivity in the PPI network despite the gene being down-regulated with no significant pathway. These observations have resulted from the fact that kidney tubular epithelial cells are pathologically exposed to massive urinary proteins in patients with glomerular diseases (*Remuzzi & Bertani, 1998*). The experimental evidence demonstrated that urinary proteins, which include albumin, are involved in the mechanism of tubulointerstitial fibrosis (*Eddy, 2004*; *Remuzzi & Bertani, 1998*). The ALB gene was down-regulated because in this bioinformatic analysis we use only the glomeruli tissues.

Although bioinformatics technology is a great method to identify the candidate genes contributing to diseases, many limitations still remain in this study. First, microarray data was downloaded from the GEO database instead of that developed by our research group. Second, the number of datasets and the sample size used in the analysis are small. Besides, the clinical data of the patients are not available, thus some confounding factors like age, sex, and renal function were not controlled in the analysis. Despite these limitations, our findings still have important implications for the molecular mechanisms of IgAN and further research is required to validate the results obtained in our study.

## CONCLUSION

In conclusion, the network analysis identified several primary genes for IgAN. Comparatively, FN1 and ALB are the most common genes among all the genes in the

three profiles. In depth functional studies on these common genes may improve our understanding of the pathological processes of IgAN. However, these findings require experimental confirmation for future use.

### Funding

This work was financially supported by the Major Research plan of the National Natural Science Foundation of China (Grant No. 91742204), international (regional) cooperation and exchange projects, (NSFC-DFG, Grant No. 81761138041), the National Natural Science Foundation of China (Grants 81470948, 81670633, 81570667), the National Key Research and Development Program (Grants 2016YFC0906103) and the National Key Technology R&D Program (Grant 2013BAI09B06, 2015BAI12B07). The funders had no role in study design, data collection and analysis, decision to publish, or preparation of the manuscript.

### Grant Disclosures

The following grant information was disclosed by the authors:
National Natural Science Foundation of China: 91742204.
International (regional) cooperation and exchange projects: 81761138041.
National Natural Science Foundation of China: 81470948, 81670633, 81570667.
National Key Research and Development Program: 2016YFC0906103.
The National Key Technology R&D Program: 2013BAI09B06, 2015BAI12B07.

### Competing Interests

The authors declare there are no competing interests.

### Author Contributions

- Mohammed Khamis Miraji and Yichun Cheng conceived and designed the experiments, analyzed the data, prepared figures and/or tables, authored or reviewed drafts of the paper, approved the final draft.
- Shuwang Ge and Gang Xu conceived and designed the experiments, approved the final draft.

### Data Availability

We used three gene expression profiles (GSE37460, GSE93798 and GSE104948) from the Gene Expression Omnibus database of NCBI.

### Supplemental Information

Supplemental information for this article can be found online at http://dx.doi.org/10.7717/peerj.7067#supplemental-information.

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
