# Peer review of "Identification of primary genes in glomeruli compartment of immunoglobulin A nephropathy by bioinformatic analysis"

_PeerJ, doi:10.7717/peerj.7067_

## Round 0.1 · original submission · Major Revisions

Your manuscript has some interesting information.

I would like to invite you to submit a revision, fully addressing the concerns of the reviewers.

·

Basic reporting

The manuscript submitted by Miraji A. et. al. reported the gene expression profiling data performed on the glomeruli of patients with IgA nephropathy and normal controls using the Gene expression Omnibus database system. Several studies earlier reported the association of these differentially expressed genes including CX3CR1 and fibronectin in case of IgA nephropathy (Romgnani, 1999, JASN). So, the gene expression profiling and the reported genes in case of IgA nephropathy may not be novel but still important at this juncture. I have a few comments.
1. They reported that total 28 genes were upregulated and 12 genes were downregulated. But the authors briefly discussed fibronectin, CD44, CX3CR1, and CCL4. Also, it is well reported these genes related to chemokines CX3CR1 and CCL4 involve in the signaling pathways. It would be interesting if they discuss and specify the name of the signaling pathway and their role(s) in the progression of IgA nephropathy. Also, they should support their findings by citing earlier published reports. For. e.g a study by Berthous F.C, 2006, Kidney Int. (not cited in the paper) reported the genetic polymorphism in the CCR5 (chemokine receptor for CCL4) in patients with IgA nephropathy and showed the genotype-phenotype correlation.
2. It is advised to explain the details of the database selection criteria used for the selection of glomeruli samples from patients with IgA nephropathy and healthy controls in the method section.
3. It is recommended to revise the language and make it crisper. For example line 130, it is mentioned: “Cellular components analysis showed that up-regulated DEGs were mainly involved in the extracellular region, extracellular space and integral component of plasma membrane”. There is no difference in the meaning of “extracellular region” and “extracellular space”. Line 134, it is mentioned, “Down-regulated DEGs were mainly involved in sequence-specific DNA binding, transcriptional activator activity and DNA binding”. “DNA binding” reads repetitious!
4. A careful proof of the manuscript is recommended to correct grammatical and typographical errors. For e.g, line 122, “The DEGs was annotated...”
5. References: be consistent throughout the m/s. a) At most of the places references are cited with the name of the authors except ref.14, line 102. b) The font style of some of the references (line 135, 318, 364) is different from others.

Experimental design

The detailed comments are mentioned in the basic reporting section. They need to specify the selection criteria used for selecting the databases for glomeruli samples from patients with IgA nephropathy and healthy controls in the method section.

Validity of the findings

See general comments for the authors.

Additional comments

The manuscript submitted by Miraji A. et. al. reported the gene expression profiling data performed on the glomeruli of patients with IgA nephropathy and normal controls using the Gene expression Omnibus database system. Several studies earlier reported the association of these differentially expressed genes including CX3CR1 and fibronectin in case of IgA nephropathy (Romgnani, 1999, JASN). So, the gene expression profiling and the reported genes in case of IgA nephropathy may not be novel but still important at this juncture. I have a few comments.
1. They reported that total 28 genes were upregulated and 12 genes were downregulated. But the authors briefly discussed fibronectin, CD44, CX3CR1, and CCL4. Also, it is well reported these genes related to chemokines CX3CR1 and CCL4 involve in the signaling pathways. It would be interesting if they discuss and specify the name of the signaling pathway and their role(s) in the progression of IgA nephropathy. Also, they should support their findings by citing earlier published reports. For. e.g a study by Berthous F.C, 2006, Kidney Int. (not cited in the paper) reported the genetic polymorphism in the CCR5 (chemokine receptor for CCL4) in patients with IgA nephropathy and showed the genotype-phenotype correlation.
2. It is advised to explain the details of the database selection criteria used for the selection of glomeruli samples from patients with IgA nephropathy and healthy controls in the method section.
3. It is recommended to revise the language and make it crisper. For example line 130, it is mentioned: “Cellular components analysis showed that up-regulated DEGs were mainly involved in the extracellular region, extracellular space and integral component of plasma membrane”. There is no difference in the meaning of “extracellular region” and “extracellular space”. Line 134, it is mentioned, “Down-regulated DEGs were mainly involved in sequence-specific DNA binding, transcriptional activator activity and DNA binding”. “DNA binding” reads repetitious!
4. A careful proof of the manuscript is recommended to correct grammatical and typographical errors. For e.g, line 122, “The DEGs was annotated...”
5. References: be consistent throughout the m/s. a) At most of the places references are cited with the name of the authors except ref.14, line 102. b) The font style of some of the references (line 135, 318, 364) is different from others.

Reviewer 2 ·

Basic reporting

no comment

Experimental design

no comment

Validity of the findings

no comment

Additional comments

In this manuscript, the authors integrated mRNA profiling studies to identify differentially expressed genes in IgAN glomeruli samples. There are some concerns before the manuscript could be published.
1. The criteria of selecting mRNA studies deposited in GEO database should be more carefully checked. As far as I am concerned, there are some additional datasets containing IgAN with more larger sample size.
2. For multiple correction in statistics, proper method should be adopted.
3. In expression data, there would be more than one probes designed for different exons in a single gene. The issue should be carefully checked for expression in a single study and in a combined analysis from multiple studies.
4. I would recommend to conduct a meta-analysis with these data instead of just checking the overlapping, and a strictly quality control should be taken before further statistics.
5. The introductions of the PPI in the METHODS section should be moved to the INTRODUCTION section.
6. It is confused that both the positive regulation of transcription from RNA polymerase II promoter and the negative regulation of transcription from RNA polymerase II promoter exist in down-regulated DEGs.
7. There are some overstatements. i.e. “Mesangial cells proliferation is a pathological feature in IgAN, thus FN1 is directly associated with the occurrence and development of primary IgAN”
8. The references should be updated, especially in the DISCUSSION section.
9. The limitations of the study design and the methodology should be discussed.

---

## Round 0.2 · Minor Revisions

It is now acceptable. However, some specific parts need minor revisions.
Thanks a lot
Cheorl-Ho Kim
Editor

·

Basic reporting

The author added a brief description of the potential signaling pathways associated with the genes that were upregulated in patients with IgA nephropathy. They did not propose the potential/possible pathway(s) related to the downregulated genes and mentioned that there was no significant pathway involved with the down-regulated genes. There are still several typographical and grammatical errors in the manuscript. For example:

a. Line 32 replace “Therefore, IgAN has been a healthy concern throughout the world” with “Therefore, IgAN has been a health concern throughout the world”
b. Line 65, replace “(AQCg and AQCp) and consistency quality control (CQCg and CQCp)” with “(AQCg and AQCp) and consistent quality control (CQCg and CQCp)”.
c. Line 138, replace “fibronetin” with “fibronectin”.
d. Line 142, correct “Therefore, the expression of FN1 gene may also influeced the..” with “Therefore, the expression of FN1 gene may also influence the..”
e. Line 154, replace “CX3CL1/CX3CR1 axis can up-regulates..” with “CX3CL1/CX3CR1 axis can up-regulate..”
f. Line 163, correct “end stage real disease” with end-stage renal disease”.
g. Line 167, correct spelling of mechanism as written “mechanism”
h. Line 1. Title replace “a nephropathy” with “A nephropathy”
i. Minor, Line 7, correct spelling of author as written “aurthor”

Experimental design

We have earlier discussed this m/s. The comments on the revised version are discussed in the basic reporting.

Validity of the findings

We have earlier discussed this m/s. The comments on the revised version are discussed in the basic reporting.

Additional comments

The author added a brief description of the potential signaling pathways associated with the genes that were upregulated in patients with IgA nephropathy. They did not propose the potential/possible pathway(s) related to the downregulated genes and mentioned that there was no significant pathway involved with the down-regulated genes. There are still several typographical and grammatical errors in the manuscript. For example:

a. Line 32 replace “Therefore, IgAN has been a healthy concern throughout the world” with “Therefore, IgAN has been a health concern throughout the world”
b. Line 65, replace “(AQCg and AQCp) and consistency quality control (CQCg and CQCp)” with “(AQCg and AQCp) and consistent quality control (CQCg and CQCp)”.
c. Line 138, replace “fibronetin” with “fibronectin”.
d. Line 142, correct “Therefore, the expression of FN1 gene may also influeced the..” with “Therefore, the expression of FN1 gene may also influence the..”
e. Line 154, replace “CX3CL1/CX3CR1 axis can up-regulates..” with “CX3CL1/CX3CR1 axis can up-regulate..”
f. Line 163, correct “end stage real disease” with end-stage renal disease”.
g. Line 167, correct spelling of mechanism as written “mechanism”
h. Line 1. Title replace “a nephropathy” with “A nephropathy”
i. Minor, Line 7, correct spelling of author as written “aurthor”

Reviewer 2 ·

Basic reporting

no comment

Experimental design

no comment

Validity of the findings

no comment

Additional comments

The authors significantly addressed the raised issues.
However, there were still some typos and grammar errors. I would like to suggest a language editing by native speakers.

---

## Round 0.3 · accepted · Accept

Your re-revision is acceptable.